# Linear-nonlinear cascades capture synaptic dynamics

**Julian Rossbroich**[1], **Daniel Trotter**[2], **John Beninger**[3], **Katalin Tóth**[3], **Richard Naud**[2,3]*

**1** Friedrich Miescher Institute for Biomedical Research, Basel, Switzerland, **2** Department of Physics, University of Ottawa, Ottawa, ON, Canada, **3** uOttawa Brain Mind Institute, Center for Neural Dynamics, Department of Cellular and Molecular Medicine, University of Ottawa, Ottawa, ON, Canada

* rnaud@uottawa.ca

**Data Availability Statement:** All data and all code relevant to the manuscript can be found upon publication on GitHub at https://github.com/neuralcodinglab/flexible-stp.

## Abstract

Short-term synaptic dynamics differ markedly across connections and strongly regulate how action potentials communicate information. To model the range of synaptic dynamics observed in experiments, we have developed a flexible mathematical framework based on a linear-nonlinear operation. This model can capture various experimentally observed features of synaptic dynamics and different types of heteroskedasticity. Despite its conceptual simplicity, we show that it is more adaptable than previous models. Combined with a standard maximum likelihood approach, synaptic dynamics can be accurately and efficiently characterized using naturalistic stimulation patterns. These results make explicit that synaptic processing bears algorithmic similarities with information processing in convolutional neural networks.

## Author summary

Understanding how information is transmitted relies heavily on knowledge of the underlying regulatory synaptic dynamics. Existing computational models for capturing such dynamics are often either very complex or too restrictive. As a result, effectively capturing the different types of dynamics observed experimentally remains a challenging problem. Here, we propose a mathematically flexible linear-nonlinear model that is capable of efficiently characterizing synaptic dynamics. We demonstrate the ability of this model to capture different features of experimentally observed data.

## Introduction

The nervous system has evolved a communication system largely based on temporal sequences of action potentials. A central feature of this communication is that action potentials are communicated with variable efficacy on short (10 ms—10 s) time scales [1–6]. The dynamics of synaptic efficacy at short time scales, or short-term plasticity (STP), can be a powerful determinant of the flow of information, allowing the same axon to communicate independent

**Funding:** This work was supported by Canadian Institutes of Health Research (CIHR):Project Grant RN38364 (RN and KT, providing salary to JB), Neurasmus program EMJMD scholarship (JR, providing salary to JR) and Natural Sciences and Engineering Research Council of Canada (NSERC) Discovery Grant 06972 (RN, providing salary to DT). The funders had no role in study design, data collection and analysis, decision to publish, or preparation of the manuscript.

**Competing interests:** The authors have declared that no competing interests exist.

messages to different post-synaptic targets [7, 8]. Properties of STP vary markedly across projections [9–11], leading to the idea that connections belong to distinct classes [12, 13] and that these distinct classes shape information transmission in vivo [14–16]. Thus, to understand the flow of information in neuronal networks, structural connectivity must be indexed with an accurate description of STP properties.

One approach to characterizing synaptic dynamics is to perform targeted experiments and extract a number of summary features. The most common feature extracted is the paired-pulse ratio [5, 17–19], which is inferred by presenting two stimulations and taking the ratio of the response to the second stimulation over the response to the first. This ratio can be used to classify a synapse as short-term depressing (STD) or short-term facilitating (STF). In addition, longer and more complex stimulation patterns suggest a variety of STP types, such as delayed facilitation onset [6], biphasic STP [20, 21] and distinct supra- and sub-linear facilitation [22]. However, without a model it is difficult to understand which observations are consistent with each other, and which come as a surprise. If it is both accurate and flexible, a model can compress the data into a small number of components.

Previous efforts have fit a mechanistic mathematical model using all available experimental data, with parameters that correspond to physical properties [23]. In this vein, the model proposed by Tsodyks and Markram captures the antagonism between transient increases in vesicle release probability and transient depletion of the readily releasable vesicle pool [11, 24, 25]. Optimizing parameter values to best fit the observed data provides an estimate of biophysical properties [26, 27]. This simple model is highly interpretable, but its simplicity restricts its ability to capture the diversity of synaptic responses to complex stimulation patterns. Complex STP dynamics rely on interactions between multiple synaptic mechanisms that cannot be described in a simplified framework of release probability and depletion. To describe the dynamics of complex synapses, the Tsodyks-Markram model therefore requires multiple extensions [23, 28], such as vesicle priming, calcium receptor localization, multiple timescales, or use-dependent replenishment [6, 29–31]. As a compendium of biophysical properties is collected, these properties become increasingly difficult to adequately characterize based on experimental data because degeneracies and over-parametrization lead to inefficient and non-unique characterization. Taken together, current approaches appear to be either too complex for accurate characterization, or insufficient to capture all experimental data.

The trade-off between a model's interpretability and its ability to espouse complex experimental data echoes similar trade-offs in other fields, such as in the characterization of the input-output function of neurons [32–37]. Taking a systems identification approach, we chose to sacrifice some of our model's interpretability in order to avoid over-parametrization and degeneracies while still capturing the large range of synaptic capabilities. Inspired by the success of linear-nonlinear models for the characterization of cellular responses [32, 33], we extended previous phenomenological approaches to synaptic response properties [3, 4, 38, 39] to account for nonlinearities and kinetics evolving on multiple time scales. The resulting Spike Response Plasticity (SRP) model captures short-term facilitation, short-term depression, biphasic plasticity, as well as sub- and supra-linear facilitation and post-burst potentiation. Using standard gradient descent algorithms, model parameters can be inferred accurately with limited amounts of experimental data. Because it combines a convolution with a nonlinear readout, our modelling framework has striking parallels with convolutional neural networks. That is, our framework suggests that synaptic dynamics can be conceptualized as extending information processing that occurs via dendritic integration, with similar information processing occurring in synapses.

## Results

### Deterministic dynamics

To construct our statistical framework, we first considered the deterministic dynamics of synaptic transmission. Our goal was to describe the dynamics of the amplitude of individual post-synaptic currents (PSCs). Specifically, a presynaptic spike train will give rise to a post-synaptic current trace, $I(t)$, made of a sum of PSCs triggered by presynaptic action potentials at times $t_j$:

$$I(t) = \sum_j \mu_j k_{PSC}(t - t_j), \tag{1}$$

where $k_{PSC}$ is the stereotypical PSC time course and $\mu_j$ is the synaptic efficacy, or relative amplitude, of the $j$th spike in the train normalized to the first spike in the train ($\mu_1 = 1$).

To begin modeling synaptic dynamics, we sought a compact description for generating $I(t)$ from the presynaptic spike train, $S(t)$. Spike trains are mathematically described by a sum of Dirac delta-functions, $S(t) = \sum_j \delta(t - t_j)$ [35]. For our purposes, we assumed the time course of individual PSCs to remain invariant through the train, but with a dynamic amplitude. To capture these amplitude changes, we introduce the concept of an efficacy train, $E(t)$, made of a weighted sum of Dirac delta-functions: $E(t) = \sum_j \mu_j \delta(t - t_j)$. The efficacy train can be conceived as a multiplication between the spike train and a time-dependent signal, $\mu(t)$, setting the synaptic efficacy at each moment of time

$$E(t) = \mu(t)S(t). \tag{2}$$

Thus the current trace can be written as a convolution of the efficacy train and the stereotypical PSC shape, $\mathbf{k}_{PSC}$: $I = \mathbf{k}_{PSC} * E$, where $*$ denotes a convolution. In this way, because in typical electrophysiological assays of synaptic properties the PSC shape ($\mathbf{k}_{PSC}$) is known and the input spike train $S(t)$ is controlled, characterization of synaptic dynamics boils down to a characterization of how the synaptic efficacies evolve in response to presynaptic spikes. Mathematically, we sought to identify the functional $\mu[S(t)]$ of the spike train $S(t)$.

Using this formalism, we aim to build a general framework for capturing synaptic efficacy dynamics. Previous modeling approaches of STP have used a system of nonlinear ordinary differential equations to capture $\mu(t)$ separated in a number of dynamic factors [4, 11, 23, 24]. Our main result is that we propose a linear-nonlinear approach inspired from the engineering of systems identification [33, 40–47] and the Spike Response Model (SRM) for cellular dynamics [34, 48, 49]. Here, the efficacies are modeled as a nonlinear readout, $f$, of a linear filtering operation:

$$\mu = \frac{1}{f(b)} f(\mathbf{k}_\mu * S + b) \tag{3}$$

where $\mathbf{k}_\mu(t)$ is the *efficacy kernel* describing the spike-triggered change in synaptic efficacy and $b$ is a baseline parameter, which could be absorbed in the definition of the efficacy kernel. The efficacy kernel can be parametrized by a linear combination of nonlinear basis functions (see Methods). Importantly, although $\mathbf{k}_\mu$ can be formalized as a sum of exponentially decaying functions, the choice of basis functions does not force a specific timescale onto the efficacy kernel. Instead, it is the relative weighting of different timescales that will be used to capture the effective timescales. In this way, while $\mathbf{k}_{PSC}$ regulates the stereotypical time-course of an isolated PSC, the efficacy kernel, $\mathbf{k}_\mu$, regulates the stereotypical changes in synaptic efficacy following a pre-synaptic action potential. The efficacy kernel can take any strictly causal form ($k_\mu(t) = 0$ for $t \in (-\infty, 0]$), such that a spike at time $t_j$ affects neither the efficacy before nor at

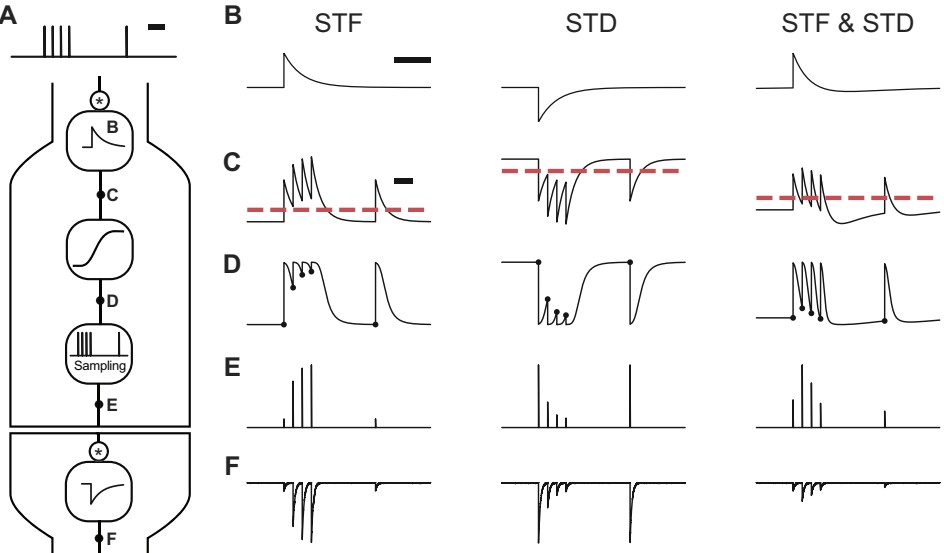

**Fig 1. The SRP model captures different types of short-term plasticity.** (**A**) The model first passes a pre-synaptic spike train through a convolution with the efficacy kernel. We illustrate three choices of this efficacy kernel: (**B**), a positive kernel for STF (left), a negative kernel for STD (middle) and one for STF followed by STD (right). After the convolution and combination with a baseline (**C**; dashed line indicates zero), a nonlinear readout is applied, leading to the time-dependent efficacy $\mu(t)$ (**D**). This time-dependent signal is then sampled at the spike times, leading to the efficacy train (**E**) and thus to the post-synaptic current trace (**F**). Scale bars correspond to 100 ms.

time $t_j$, but only after $t_j$. Here we call the 'potential efficacy' the result of the convolution and baseline, $\mathbf{k}_\mu * S + b$, before taking a sigmoidal nonlinear readout. Although some early studies have used a linear readout [4], synaptic dynamics invoke mechanisms with intrinsic nonlinearities, like the saturation of release probability or the fact that the number of vesicles cannot be negative. The readout, $f(\cdot)$, will capture the nonlinear progression of PSC amplitudes in response to periodic stimulation. The factor $f(b)^{-1}$ was introduced because we consider the amplitudes normalized to the first pulse, replaceable by an additional parameter when treating non-normalized amplitudes. This version of the deterministic SRP model, can capture different types of STP by changing the shape of the efficacy kernel.

**Short-term facilitation and depression.** To show that the essential phenomenology of both STF and STD can be encapsulated by an efficacy kernel $\mathbf{k}_\mu$, we studied the response to a burst of four action potentials followed by a delay and then a single spike and compared responses obtained when changing the shape of the efficacy kernel (Fig 1A). For simplicity, we considered $\mathbf{k}_\mu$ to be a mono-exponential decay starting at time 0. When the amplitude of this filter is positive (Fig 1B, left), a succession of spikes leads to an accumulation of potential efficacy ($\mathbf{k}_\mu * S + b$, Fig 1C, left). After the sigmoidal readout (Fig 1D, left) and sampling at the spike times, the efficacy train (Fig 1E, left) and the associated current trace (Fig 1F, left) showed facilitation. Choosing a negative amplitude (Fig 1B, middle) gave rise to the opposite phenomenon. In this case, the succession of spikes gradually decreased potential efficacy ($\mathbf{k}_\mu * S + b$, Fig 1C, middle). Following the sigmoidal readout (Fig 1D, middle) the efficacy train (Fig 1E, middle) and the resulting current trace (Fig 1F, middle) showed STD dynamics. Conveniently, changing the polarity of the efficacy kernel controls whether synaptic dynamics follow STF or STD.

At many synapses, facilitation apparent at the onset of a stimulus train is followed by depression, a phenomenon referred to as biphasic plasticity [20, 21, 50]. To model this biphasic

plasticity in our framework, we considered an efficacy kernel consisting of a combination of two exponential-decays with different decay timescales and opposing polarities. By choosing the fast component to have a positive amplitude and the slow component to have a negative amplitude (Fig 1B, right), we obtained a mixture between the kernel for STF and the kernel for STD. Under these conditions, a succession of spikes creates an accumulation of potential efficacy followed by a depreciation ($\mathbf{k}_u * S + b$, Fig 1C, right). Once the sigmoidal readout was performed (Fig 1D, right), the efficacy train (Fig 1E, right) and the resulting PSC trace (Fig 1F, right) showed facilitation followed by depression. Thus, the model captured various types of STP by reflecting the facilitation and depression in positive and negative components of the efficacy kernel, respectively.

**Sublinear and supralinear facilitation.** The typical patterns of facilitation and depression shown in Fig 1 are well captured by the traditional Tsodyks-Markram (TM) model [24–26]. This model captures the nonlinear interaction between depleting the readily releasable pool of vesicles (state variable $R$) and the probability of release (state variable $u$; see Methods for model description). We, therefore, asked whether our modeling framework could capture experimentally observed features that require a modification of the classical TM model. While previous work has extended the TM model for use-dependent depression [29] and receptor desensitization [23], we considered the nonlinear facilitation observed in mossy fiber synapses onto pyramidal neurons (MF-PN) in response to a burst of action potentials (Fig 2A). In these experiments, the increase of PSC amplitudes during the high-frequency stimulation was nonlinear. Interestingly, the facilitation was sublinear at normal calcium concentrations (2.5 mM extracellular $[Ca^{2+}]$), but supralinear in physiological calcium concentrations (1.2 mM extracellular $[Ca^{2+}]$) [22] (Fig 2B). The supralinearity of STF observed in 1.2 mM $[Ca^{2+}]$ was caused by a switch from predominantly univesicular to predominantly multivesicular release. In contrast, multivesicular release was already in place in 2.5 mM $[Ca^{2+}]$, and the facilitation observed under these conditions can be solely attributed to the recruitment of additional neurotransmitter release sites at the same synaptic bouton [22]. These two mechanisms, by which MF-PN synapses can facilitate glutamate release, arise from complex intra-bouton calcium dynamics [30, 51, 52], which lead to gradual and compartmentalized increases in calcium concentration. Consistent with the expectation that these two modes could lie on the opposite sides of the inverse-parabolic relationship between coefficient of variation (CV) and mean, normal calcium was associated with a gradual decrease of CV through stimulation, while

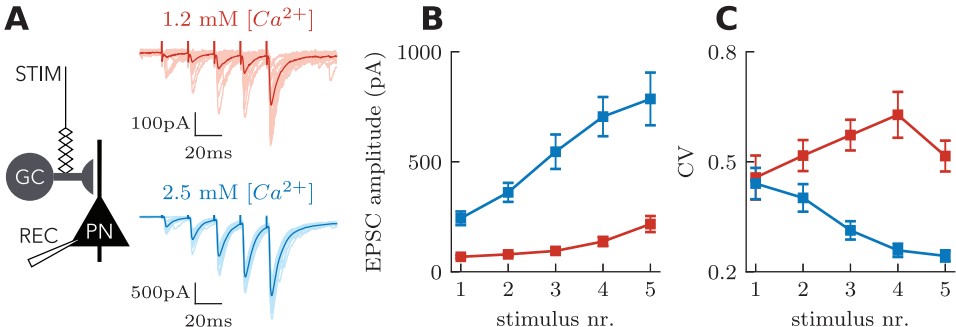

**Fig 2. Effects of extracellular calcium concentration on STP dynamics at hippocampal mossy fiber synapses. A** Mossy fiber short-term facilitation in 1.2 mM (red) and 2.5 mM (blue) extracellular $[Ca^{2+}]$. PSCs recorded from CA3 pyramidal cells in response to stimulation of presynaptic mossy fibers (50 Hz, 5 stimuli). **B** PSC peak amplitudes as a function of stimulus number. The time course of facilitation varies dependent on the initial release probability. **C** The coefficient of variation (CV), measured as the standard deviation of PSCs divided by the mean, is increased in 1.2 mM extracellular $[Ca^{2+}]$. Data redrawn from Chamberland et al. (2014) [22].

physiological calcium was associated with an increase of CV (Fig 2C). Perhaps because the TM model was based on experiments at 2 mM calcium concentration, the model emulates sublinear facilitation. Supralinear facilitation is not possible in the original structure of the model (Fig 3C), as can be verified by mathematical inspection of the update equations (see Methods).

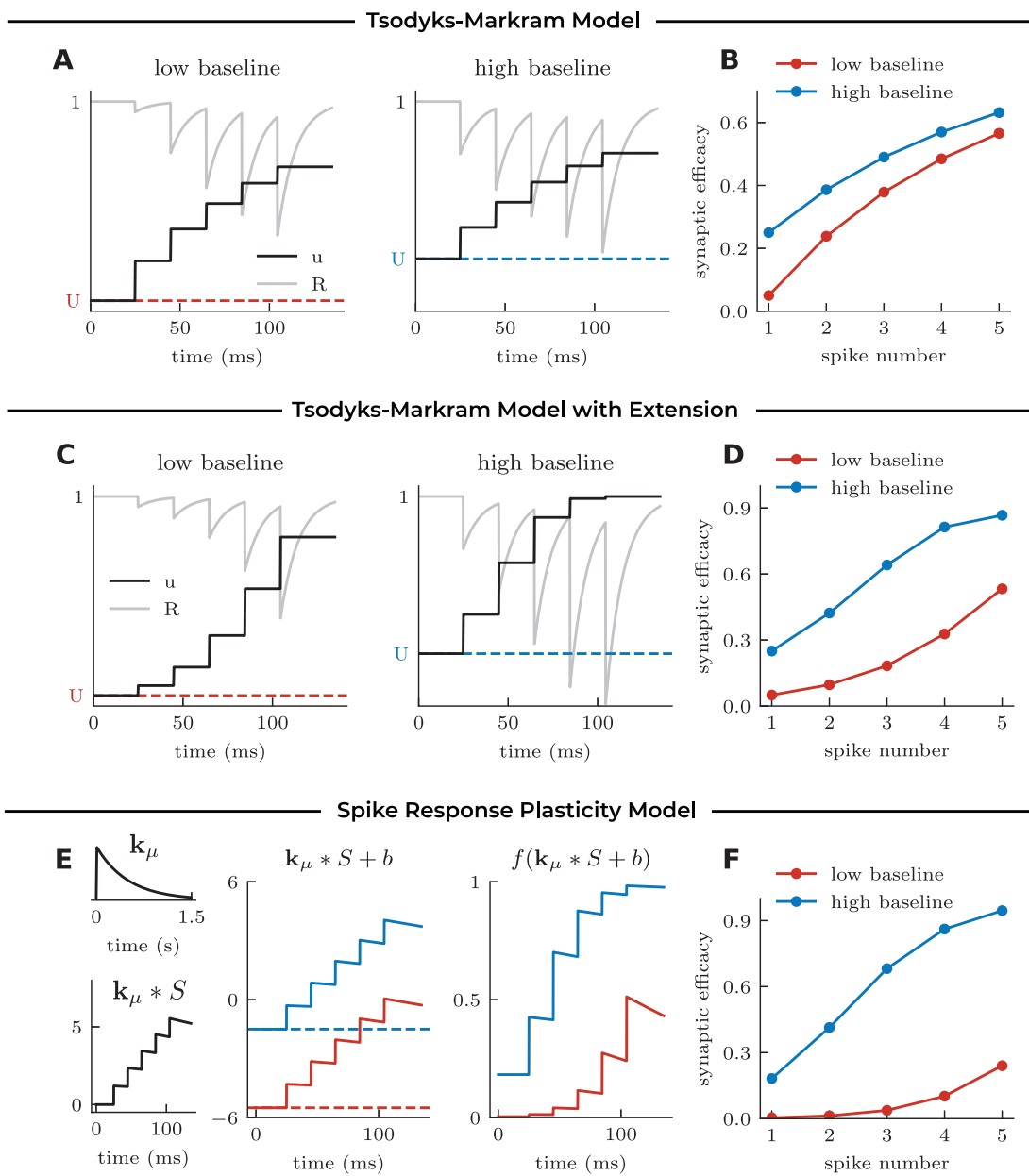

**Fig 3. Modeling sublinear and supralinear facilitation through changes in the baseline parameter. A** Mechanism of the classic TM model [24–26], illustrated in response to 5 spikes at 50 Hz for different values of the baseline parameter $U$. **B** Synaptic efficacy $uR$ at each spike according to the classic TM model. Facilitation is always restricted to sublinear dynamics. **C** Mechanism and **D** Synaptic efficacy $uR$ at each spike according to the extended TM model (see Methods). Choosing the baseline parameter $U$ sufficiently small allows for supralinear facilitation. **E** Mechanism of the SRP model, illustrated for two different values of the baseline parameter $b$, with the same synaptic efficacy kernel $k_\mu$ (left). Changing the baseline parameter $b$ leads to a linear displacement of the filtered spike train $k_\mu * S + b$ (middle), which causes a shift from sub- to supralinear dynamics after the nonlinear readout $f(k_\mu * S + b)$. **F** Resulting synaptic efficacy at each spike according to the SRP model. Changing the baseline parameter causes a switch from sublinear to supralinear facilitation, as observed experimentally in response to varying extracellular $[Ca^{2+}]$ (see Fig 2).

Hence the TM model must be modified to capture the supralinear facilitation typical of experimental data at physiological calcium concentrations.

To extend the TM model to account for supralinear facilitation, we considered a small modification to the dynamics of facilitation without adding a new dynamic variable (Fig 3A), although supralinear facilitation can be achieved with an additional state variable. This modification allows the facilitation variable of the TM model $u$ to increase supralinearly when $u$ is small, and sublinearly when $u$ is large (see Methods). By lowering the baseline facilitation parameter $U$, the extended TM model switches from sublinear facilitation to a supralinear facilitation (Fig 3D). We thus have shown that a modification to the set of equations for the TM model is required to present supralinear facilitation and capture the experimentally observed facilitation at physiological calcium.

In contrast, for the linear-nonlinear model framework, the switch from sublinear to supralinear facilitation does not require a modification to the equations. We can change sublinear facilitation into supralinear facilitation by lowering the baseline parameter without changing the efficacy kernel. When the baseline parameter is high, a facilitating efficacy kernel is likely to hit the saturating, sublinear, part of the nonlinear readout (Fig 3E). When the baseline parameter is low, the same facilitating efficacy kernel can recruit the onset of the nonlinearity, which gives rise to supralinear facilitation (Fig 3F). Thus, changes in extracellular calcium concentration are conveniently mirrored by the modification of a baseline parameter in the SRP model. Later in this manuscript, we expand the modelling framework to account for probabilistic synaptic transmission and demonstrate that the modification of the baseline parameter similarly explains the experimentally observed changes in CV.

**Facilitation latency.**   Next we illustrate the role of the efficacy kernel to generalize to the multiple timescales of STP without requiring a change in the structure of the model. As an illustrative example, we focused on a particular synapse showing facilitation latency [6]. In mossy fiber synapses onto inhibitory interneurons, the facilitation caused by a burst of action potentials increases during the first 2 seconds after burst (Fig 4A). This delayed facilitation cannot be captured by the classical TM model because facilitation is modeled as a strictly decaying process and the experimental data show that facilitation increases during the first 1-2 seconds following a burst. Adding to this model a differential equation for the slow increase of facilitation is likely sufficient to capture facilitation latency, but this modification is considerable.

In the linear-nonlinear framework, one could capture the facilitation latency by modifying the shape of the efficacy kernel. An efficacy kernel with a slow upswing (Fig 4B), once convolved with a burst of action potentials followed by a test-pulse (Fig 4C) will produce a delayed increase in synaptic efficacy (Fig 4D) and match the nonlinear increase in facilitation with the number of stimulation spikes. Without automated fitting of the kernel to the data, a simple change to the efficacy kernel captures facilitation latency. The same model also captured the potentiation of amplitudes as a function of the number of action potentials in the burst (Fig 4E). Thus, provided that the efficacy kernel is parameterized with basis function spanning a large part of the function space, the SRP model can aptly generalize to STP properties unfolding on multiple timescales.

## Stochastic properties

Synaptic transmission is inherently probabilistic. The variability associated with synaptic release depends intricately on stimulation history, creating a complex heteroskedasticity. Such changes in variability may be a direct reflection of history-dependent changes in amplitudes. Although a fixed relationship between the mean amplitude and the variance of synaptic

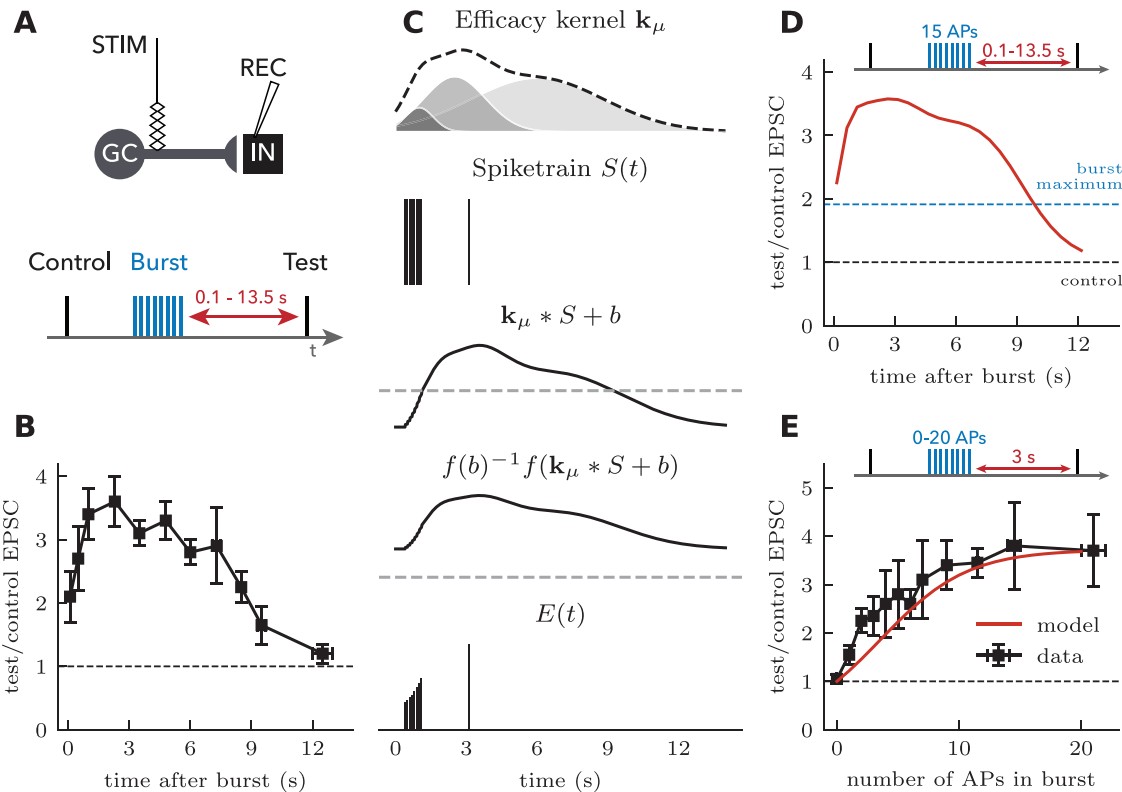

**Fig 4. Post-burst facilitation captured by a delayed facilitation kernel. A** Experimental setup and **B** measurement of post-burst facilitation in CA3 interneurons (redrawn from Ref. [6]). **C** Synaptic plasticity model. A delayed facilitation kernel was chosen as the sum of three normalized Gaussians with amplitudes {125, 620, 1300}, means {1.0, 2.5, 6.0} s and standard deviation {0.6, 1.3, 2.8} s. The spike train (8 spikes at 100 Hz followed by a test spike) is convolved with the delayed facilitation kernel. A nonlinear (sigmoidal) readout of the filtered spike train leads to synaptic efficacies. Dashed lines indicate zero. **D** Efficacies of test spikes in the synaptic plasticity model as a function of the number of action potentials in the preceding burst. **E** Synaptic efficacy of test spikes (3 s after a single burst at 160 Hz) as a function of the number of action potentials (APs). Data redrawn from Ref. [6].

responses could be expected if the only source of variability was a fixed number of equal-sized vesicles being randomly released with a given probability (a binomial model) [53], the variability should also depend on the dynamics of both the changing number of readily releasable vesicles and the changing probability with which they release [54]. In addition, other sources of variability are present such as the mode of release [55] or the size of vesicles [56, 57]. Fig 2C illustrates heteroskedasticity observed experimentally whereby the variability increases through a stimulation train but only for the physiological calcium condition. To capture these transmission properties, we established a stochastic framework. Since the mechanisms underlying the dynamics of the variability of synaptic release are not known, we first constructed a flexible but complex model, and considered simplifications as special cases.

In the previous section, we treated the deterministic case, which corresponds to the average synaptic efficacies. We next considered a sample of synaptic efficacies to be a random variable such that the $j$th spike was associated with the random variable $Y_j$. Its mean is given by the linear-nonlinear operation:

$$\langle Y_j \rangle \equiv \mu_j = \frac{1}{f(b)} f(\mathbf{k}_\mu * S(t_j) + b). \tag{4}$$

In this way, the current trace is made of PSCs of randomly chosen amplitudes whose average pattern is set by the efficacy kernel: $I(t) = \sum_j y_j \mathbf{k}_{PSC}(t - t_j)$, where $y_j$ is an instance of $Y_j$. Sampling from the model repeatedly will produce slightly different current traces, as is typical of repeated experimental recordings (Fig 2A).

To establish stochastic properties, we had to select a probability distribution for the synaptic efficacies. Previous work has argued that the quantal release of synaptic vesicles produces a binomial mixture of Gaussian distributions [53, 58]. There is substantial evidence, however, that releases at *single synapses* are better captured by a mixture of skewed distributions such as the binomial mixture of gamma distributions [56, 59]. Such skewed distributions are also a natural consequence of Gaussian-distributed vesicle diameters and the cubic transform of vesicle volumes [57]. For multiple synaptic contacts, release amplitudes should then be captured by a weighted sum of such binomial mixtures, a mixture of mixtures as it were. Indeed, a binomial mixture of skewed distributions has been able to capture the stochastic properties of PSC amplitudes from multiple synaptic contacts [27, 60], but only under the assumption that each synapse contributes equally to the compound PSC. Together, these considerations meant that for a simple parameterization of the random process, we required a skewed distribution whose mean and standard deviation could change during the course of STP.

Following prior work [56, 60], we chose to focus on gamma-distributed PSCs:

$$p(y_j|S, \theta) = g(y_j|\mu_j, \sigma_j), \tag{5}$$

where $g(y|\mu, \sigma)$ is the gamma distribution with mean, $\mu$, and standard deviation, $\sigma$. Here we assume statistical independence of successive response, $p(y_j, y_{j-1}|S, \theta) = p(y_j|S, \theta)p(y_{j-1}|S, \theta)$. The mean is set by the linear-nonlinear operation in Eq 4 and the standard deviation is set by a possibly distinct linear-nonlinear operation:

$$\sigma_i = \sigma_0 f(k_\sigma * S_i + b_\sigma), \tag{6}$$

where we introduced a baseline parameter, $b_\sigma$ and another kernel, $k_\sigma$, for controlling the standard deviation. We call this time-dependent function, the variance kernel. The factor $\sigma_0$ is introduced to scale the nonlinearity $f$ appropriately, but could be omitted if data has been standardized. In this framework, some common statistics have a simple expression in terms of model parameters. This is the case for the stationary CV. Since we are considering filters decaying to zero after a long interval and amplitudes normalized to the responses after long intervals, we have for the first pulse CV $= \sigma_1/\mu_1 = \sigma_0 f(b_\sigma)$.

This stochastic model has two important special cases. The first is the case of constant variance, which is obtained by setting the variance kernel to zero. In that case the CV of releases will be inversely proportional to the mean given in Eq 4, and thus in agreement with experimental data in 2.5 mM [Ca$^{2+}$] (Fig 2C). The other case corresponds to variability that is proportional to the mean. In this second case, we assume that the dynamics of variability follows the dynamics of the mean amplitude. For this, we set $\mathbf{k}_\sigma = \mathbf{k}_\mu$. Although both mean and variance were modeled with the same kernel, different baseline parmeters can give rise to different dynamics of the CV. Both simplifications are of interest because they drastically reduce the number of parameters in the model.

The properties of this choice of probability distribution are illustrated in Fig 5. Using a depressing kernel, Fig 5 depicts the effect of choosing a variance kernel with positive, negative and zero amplitude (Fig 5A). These kernel choices show that the model can capture both increases and decreases of variability, although an increase in variability during STD is generally observed [54, 61]. The temporal profile of the variance kernel determines the time-dependent changes in variance. For simplicity, we chose an exponential decay with a relaxation time

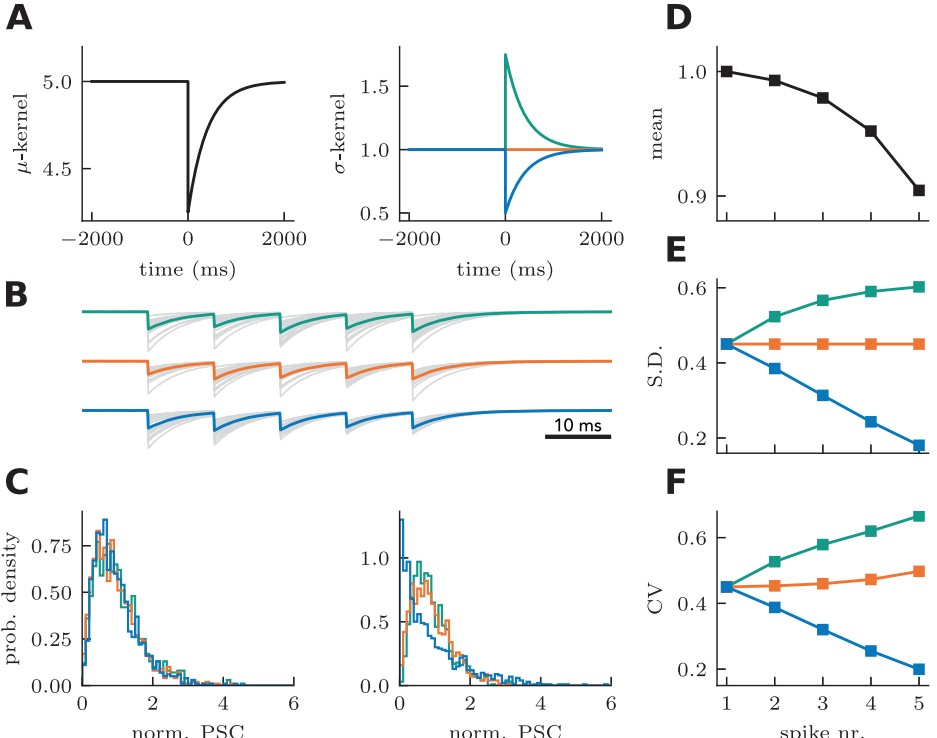

**Fig 5. Capturing heteroskedasticity with a two-kernel approach. A** The $\mu$-kernel regulating the dynamics of the mean amplitude is paired with a $\sigma$-kernel regulating the dynamics of the variance. Three $\sigma$-kernels are shown: a variance increasing (teal), a variance invariant (orange) and a variance decreasing (blue) kernel. **B** Sample PSC responses to a spike train generated from the three $\sigma$-kernels (gray lines) along with the associated mean (full lines). **C** Probability density function of the amplitude of the first (left) and last (right) pulse. **D** The mean amplitude is unaffected by different $\sigma$-kernels. **E** The standard deviation is either increasing (teal), invariant (orange) or decreasing (blue), consistent with the polarity of the $\sigma$-kernel. **F** The coefficient of variation results from a combination of $\mu$ and $\sigma$ kernel properties.

scale equal to that of the efficacy kernel. The kernel amplitude and baseline were chosen to match experimental observations at STD synapses (CV increasing from a little less than 0.5 to almost 1 after 5 pulses [54]). With these modeling choices, we simulated the probabilistic response to input trains (Fig 5B, 5 spikes, 100 Hz). The model with positive $\sigma$-kernel shows a progressive increase of trial-to-trial variability. Conversely, the model with a negative $\sigma$-kernel displays the opposite progression, as can be observed by comparing the probability distribution of the first and the last response (Fig 5C). The average response follows precisely the same STD progression (Fig 5D), despite drastically different progression of standard deviation (Fig 5E) and CV (Fig 5F). Thus gamma-distributed amplitudes with dynamic variance can capture multiple types of heteroskedasticity.

Next we asked if the model could capture the striking changes in heteroskedasticity observed in MF-PN synapses (Fig 2C). In this case, decreasing the extracellular concentration of calcium not only changed the averaged-response progression from sublinear to supralinear (Fig 2B), but also changed the CV progression from strongly decreasing to strongly increasing (Fig 2C, [22]). Fig 6 shows that changing the $\mu$-kernel baseline in a model with facilitating standard deviation can reproduce this phenomenon. Here, as in the deterministic version of the model, the change in baseline changes the progression of efficacies from sublinear to supralinear (Fig 6A–6D). These effects are associated with changes in variances that are sublinear and supralinear, respectively (Fig 6E). In the model with a low baseline (red curve in Fig 6),

                          

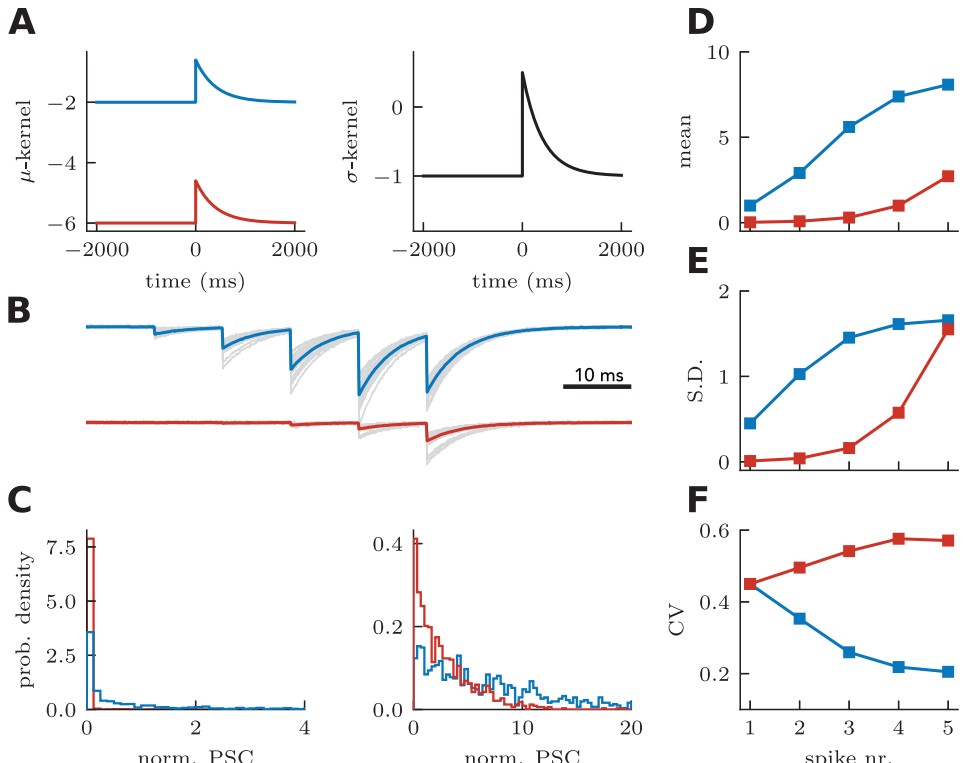

**Fig 6. Capturing effect of external calcium concentration on coefficient of variation through baseline of $\mu$-kernel.** **A** Comparing facilitating $\mu$-kernels with high (blue) and low (red) baseline but fixed, $\sigma$-kernel. **B-F** as in Fig 5. The coefficient of variation increases with pulse number for the low baseline case, but decreases with pulse number for the high baseline case.

the variance increases more quickly than the efficacy, leading to a gradual increase in CV. Despite the fact that the variance increases for both cases (Fig 6E), only the model with sublinear increase in efficacy displays a decreasing CV. We conclude that, by controlling a baseline parameter, the model can capture both the change from sublinear to supralinear facilitation and the change in heteroskedasticity incurred by a modification of extracellular calcium concentration.

## Inference

Thus far, we have illustrated the flexibility of the SRP framework for qualitatively reproducing a diversity of notable synaptic dynamics features. Next we investigated the ability of this framework to capture synaptic dynamics quantitatively. As in the characterization of cellular dynamics [62], a major impediment to precise characterization is parameter estimation. As efficient parameter inference largely depends on the presence of local minima, we first investigated the cost function landscape for estimating model parameters.

We have developed an automatic characterization methodology based on the principle of maximum likelihood (see Methods). Given our probabilistic model of synaptic release, we find optimal filter time-course by iteratively varying their shape to determine the one maximizing the likelihood of synaptic efficacy observations. The method offers a few advantages. First, the method is firmly grounded in Bayesian statistics, allowing for the inclusion of prior knowledge and the calculation of posterior distributions over the model parameters [26, 60]. Second, although targeted experiments can improve inference efficiency, our approach does not rely

                          

on experimental protocols designed for characterization. Naturalistic spike trains recorded in-vivo [30, 63], Poisson processes or other synthetic spike trains can be used in experiments to characterize synaptic dynamics in realistic conditions.

We treat the number of basis functions as well as the timescale (or shape) of the basis functions for efficacy and variance kernels as meta-parameters. Such meta-parameters are considered part of the fitting procedure, rather than a salient characteristic of a mechanistic model. We emphasize this point because, although we have parametrized the efficacy kernel as a sum of exponential decays, each characterized by a specific timescale (see Methods), we do not expect that any of these timescales match the timescale of a specific biological mechanism. One reason for this comes from the fact that it is possible to capture reasonably well a mono-exponential decay with a well chosen bi-exponential decay. Thus, a single biological timescale can be fitted by the appropriate combination of two timescales. Together, some heuristics can be applied as to the number and the choice of timescale that we expect to see in a particular system (e.g. timescales longer than 1 min would be long-term plasticity), but the choice of meta-parameters should be guided by the properties of statistical inference: choosing either a small number of well-spaced timescales to avoid overfitting, or a very large number of time-scales so as to exploit the regularizing effect of numerous parameters [35, 64, 65].

To test the efficiency of our inference method, we generated an artificial Poisson spike train with 4000 spikes at an average firing rate of 10 Hz and used this spike train to generate surrogate synaptic efficacy data using our SRP model (Fig 7A and 7B). We then asked if our inference method identified the correct parameters and whether local minima were observed. Instead of the case where the filters are described by a combination of nonlinear basis functions, we considered only one basis function, a mono-exponential decay, with its decay time constant known. In cases where the time constant is unknown, one would fit the coefficient of a combination of nonlinear basis functions, as is typical in other linear-nonlinear models [32, 34, 66, 67]. Using a long stimulus train, the likelihood function appeared convex over a fairly large range of parameter values, as no local minima were observed (Fig 7C–7F). The slanted elongation of likelihood contour indicates a correlation or anti-correlation between parameter estimates. Not surprisingly, we found that the estimates of baseline and scale factor of the $\sigma$-kernel are anti-correlated (Fig 7D), while on the other hand the estimates of filter amplitudes for efficacy and variance show a correlation (Fig 7C). To test how many training spikes would be needed for accurate parameter inference, we simulated Poisson spike trains of different lengths and used gradient descent on the likelihood function to infer all model parameters (see Methods). We found that the parameter estimates matched closely the parameters used to simulate the responses after 100-200 spikes (Fig 7G and 7H), with more training spikes leading to better parameter estimation. The relationship between error in parameter estimation and training size is such that for large training sets the percent error goes to zero (Fig 7G and 7H). Using a separate artificial Poisson input for testing the predictive power of the model, we calculated the mean squared error between the inferred and true model (Fig 7I). The prediction error of the inferred model almost matched that of the true model, even if inference was based on less than 100 spikes. We conclude that maximum likelihood applied to surrogate data is able to characterize the model efficiently and accurately, and that, for simple filters, the landscape is sufficiently devoid of local minima to allow efficient characterization.

## Model validation on mossy fiber synapses

Having established a method to infer model parameters, we now fit the model to experimental data and evaluate its accuracy for predicting the PSP amplitude to stimulation protocols that were not used for training. Furthermore, to serve as a benchmark, we will compare predictions

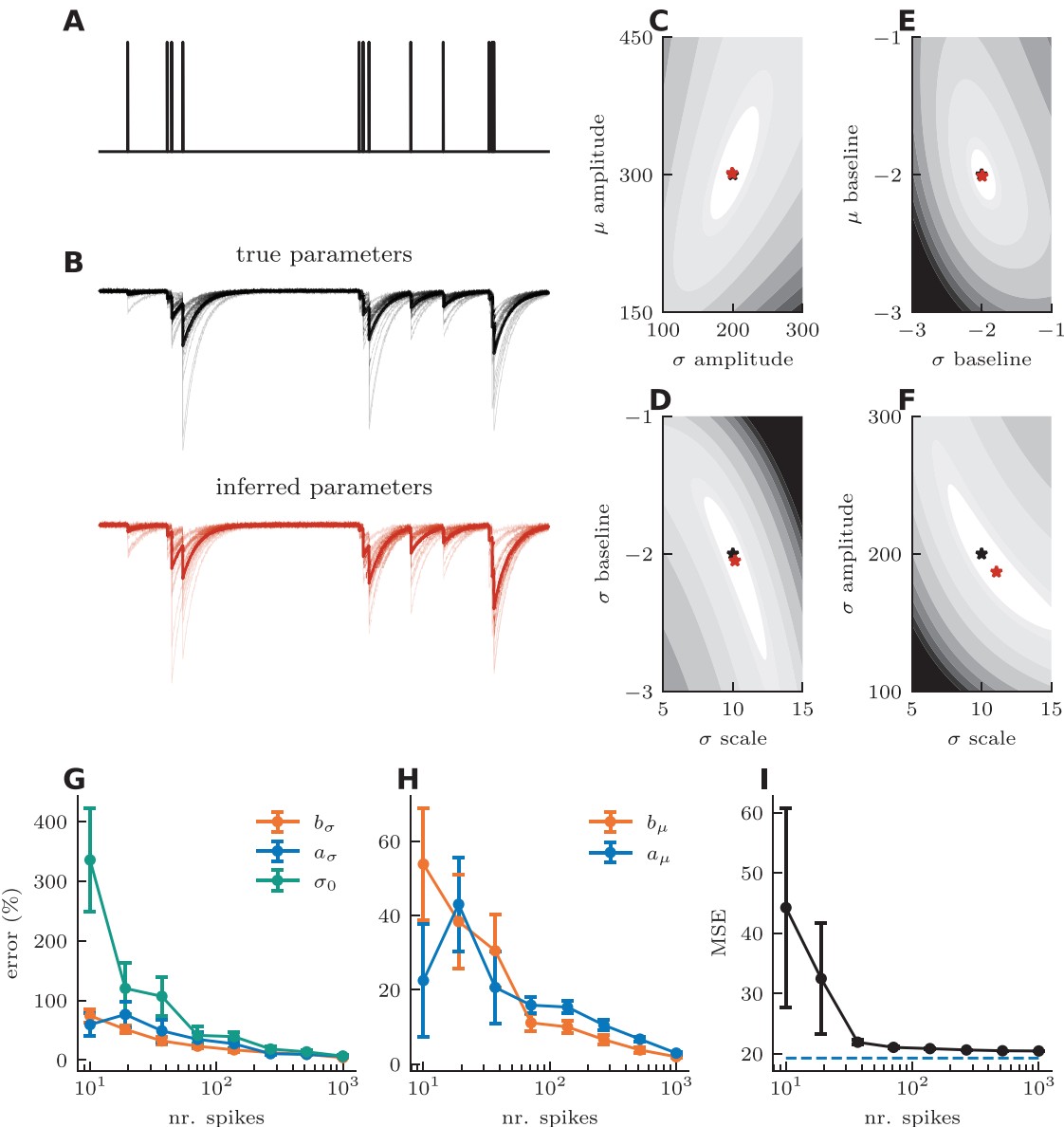

**Fig 7. Statistical inference of kinetic properties on surrogate data. A** Simulated Poisson spike trains mark pre-synaptic stimulation. **B** Simulated post-synaptic currents of the spike train in A for independent sampling (thin black lines) and mean efficacy (thick black line) of the true parameter set (top) and of the inferred parameter set (bottom). **C–F** Negative log-likelihood landscape, true parameters (black stars) and function minima (red stars) as a function of **C** $\mu$- and $\sigma$-kernel amplitudes, **D** $\sigma$ baseline and scaling factor, **E** $\mu$ and $\sigma$ baseline and **F** $\sigma$ scaling and amplitude. **G** Average $\sigma$ parameter errors as function training size. **H** average $\mu$ parameter errors as a function of training size (right). **I** Mean square error (MSE) of the inferred and model on an independent test set as a function of training size. Dashed line is MSE between independent samples of the true parameter set.

from the SRP model with those of the TM model. To do this, we used data from the mossy fiber synapse where a total of 7 different stimulation protocols were delivered and the resulting PSP amplitude were recorded: 10x100 Hz (Fig 8A), 10x20 Hz, 5x100 Hz + 1x20 Hz, 5x20 Hz + 1x100 Hz, 5x100 Hz + 1x10 Hz, 111 Hz and an in vivo recorded spike train from dentate gyrus granule cells. This experimental data was acquired at 1.2 mM extracellular [Ca$^{2+}$] in P17—25 male rats (See Ref. [30] for the complete experimental protocol).

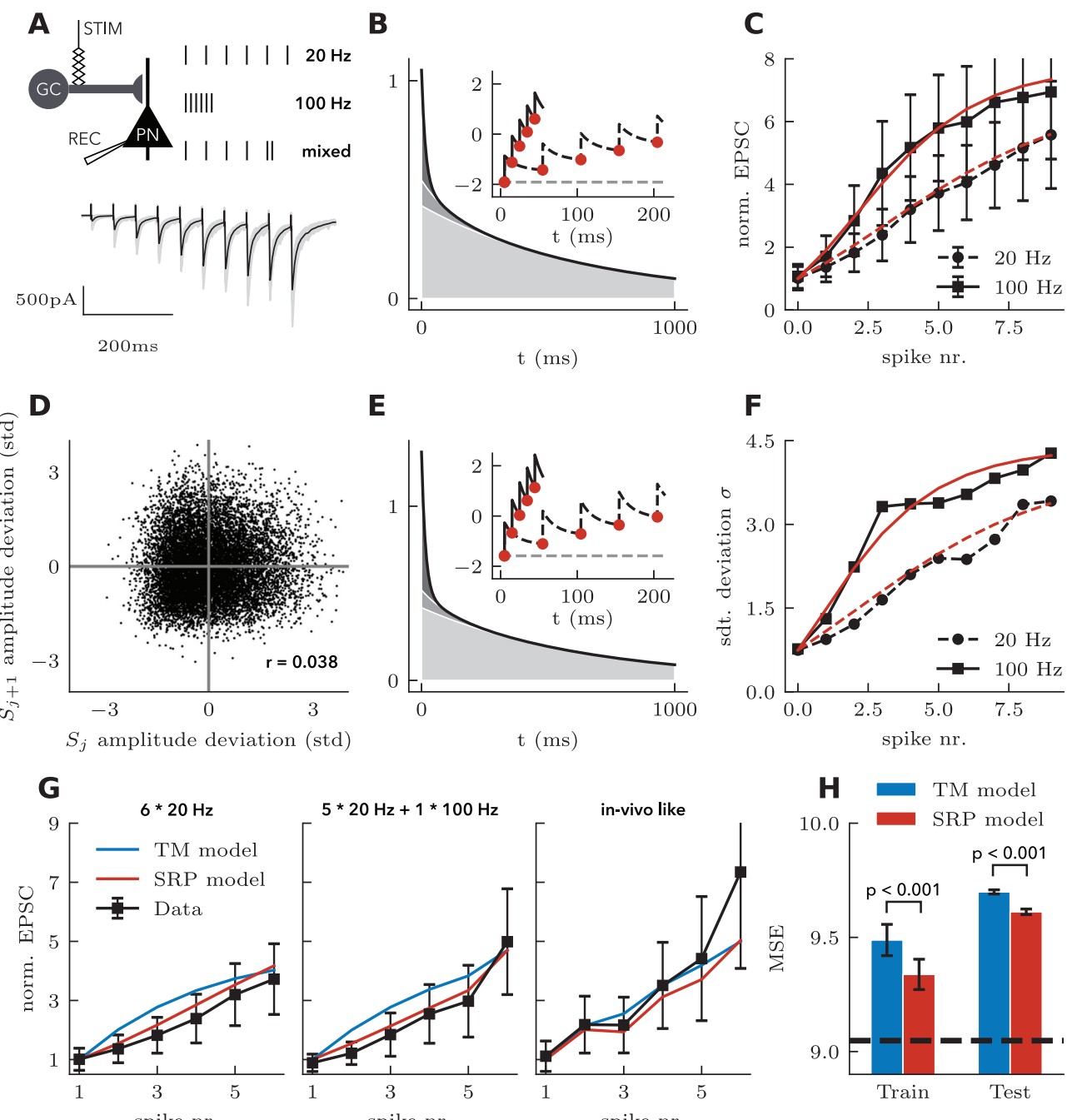

**Fig 8. Experimental validation of the SRP model. A** Experimental schematic (top) and representative PSCs recorded from CA3 pyramidal cells in response to stimulation of mossy fibers (bottom). **B** Optimal efficacy kernel (black line) is made of the combination of three exponentially decaying functions (shades of gray) with time constants $\tau = [15, 100, 650]$ ms. Inset shows the quantity $\mathbf{k_\mu} * S + b_\mu$ in response to 100 Hz (full black line) and 20 Hz (dashed black line) train. Circles indicate times where the nonlinear readout is taken and the dashed gray line indicates the baseline. **C** Normed PSC for SRP model (red lines) and data (black lines) for the regular 20 Hz (dashed lines) and 100 Hz (full lines) protocols. **D** Experimental PSC amplitude deviation (difference between an observation and its trial average) against the previous PSC amplitude deviation. **E** Optimal $\sigma$-kernel and illustration of $\mathbf{k_\sigma} * S + b_\sigma$. **F** As in **C** but showing the standard deviation. **G** Predictions of stimulation protocols held out from training. TM model (blue) and SRP model (red) predictions are shown with data (black) for the 20 Hz stimulation (left), 5x20 Hz+1x100 Hz stimulation (center) and in-vivo like stimulation (right). **H** Mean squared error (MSE) of models (bars) and variance of the data (dashed line), averaged across stimulation protocols.

Before assessing prediction accuracy, we scrutinized the model parameters fitted to all of the protocols. The optimal SRP model for this synapse had a slightly negative baseline ($b_\mu = -1.91$) and a net positive efficacy kernel which extended on multiple timescales (Fig 8B, $\theta_\mu = [7.6, 11.8, 277.0]$ for three exponential decays with time constants $\tau = [15, 100, 650]$ ms). This captures well the fact that these synapses are known to be facilitating and that multiple timescales of facilitation have been reported [22]. These parameters reproduced perfectly the nonlinear progression of PSC amplitude in response to 20 Hz and 100 Hz train (Fig 8C).

We have also validated one of the assumptions of the stochastic model, the independence of variability through subsequent sampling (Eq 5). To test this, we calculated the noise correlation across subsequent stimulation times. Fig 8D shows the deviation around the trial-averaged amplitude for one stimulation time against the deviation around the trial-averaged amplitude for the next stimulation time. Across all such amplitude pairs in the data ($n = 12040$), we found a small, but significant correlation ($r = 0.04$, $p < 0.001$). Based on the small correlation coefficient, we concluded that the effect of previous stimulation on the variability of response amplitudes is negligible and thus the model assumptions hold.

We then considered the $\sigma$-kernel found by the fitting method to capture changes in response variability. The optimal kernel was very similar (Fig 8E) to the optimal $\mu$-kernel. Both were starting from a slightly negative baseline and were made of multiple timescales, composed mostly of the fastest and slowest timescale ($b_\sigma = -1.59$, $\theta_\sigma = [11.9, 10.1, 271.6]$ for exponential decays with time constants $\tau = [15, 100, 650]$ ms). These allowed the SRP model to adequately capture the nonlinear progression in PSC variability through the stimulation of 20 Hz and 100 Hz trains (Fig 8F).

Next, we separated the data into training and test sets and only optimized the model parameters on the training set. To separate the data, we held out the data from one stimulation protocol and predicted its responses using parameters optimized on all of the other protocols. We repeated this procedure 7 times, holding out each stimulation protocol, and performed the same model optimization for both SRP and TM models. Fig 8G shows a subset of model predictions compared with observed mean amplitude. Consistent with the fact that the TM model cannot capture the supralinear increase observed after the first few stimulations at high frequency (Fig 3), the SRP model systematically outperformed the TM model for the prediction the first few stimulations. In addition, the in-vivo-like stimulation pattern was well captured by the SRP model, except for the last stimulation time that both SRP and TM models failed to predict.

To test whether the SRP model would consistently outperform the TM model, we implemented a bootstrapping procedure with 20 randomly re-sampled subsets of the data. To obtain each subset, we randomly excluded 20% of traces from every stimulation protocol. For each subset of data, we then iteratively held out each stimulation protocol, as described in the previous paragraph. This procedure results in a total of 7 TM and SRP model fits (each stimulation protocol withheld) for each of the 20 bootstrap iterations. To quantify the prediction accuracy across all held out protocols, we calculated the mean squared error (MSE). Like all metrics, the MSE weighs some features of the response more than others. Here, since later stimulations in a train are generating larger amplitudes and, therefore, larger errors, the later stimulations are weighted more than the first stimulations. Since the TM model is systematically worse on the stimulations early in the train (in part because the TM model uses MSE for parameter inference), this metric should favour the TM model. We found that, using a metric favourable to the TM model, the SRP model was more accurate in capturing both training data (paired sample t-test, $T = 45.5$ $p < 0.001$) and held out testing data (paired sample t-test, $T = 10.5$ $p < 0.001$), achieving a root mean squared error of 9.6 (Fig 8H) across all stimulation protocols, only slightly above the MSE due to intrinsic variability (dashed line in Fig 8H). The small

increase in test error from the training error indicated that some overfitting may be present in both models. Since the SRP model has more parameters (8 parameters in the SRP model with 3 basis functions versus 4 parameters in the TM model), overfitting can account for its better training error but not for the better test error. Together, we found that the SRP model predicts the response to novel stimulation patterns with high accuracy, and outperforms the TM model.

## Relation to generalized linear models

We have shown that, in one situation, the likelihood landscape appears devoid of local minima, but is this always the case? Without additional restrictions on the model described in the previous section, it is unlikely that the likelihood would be always convex. However, with some simplifications, the model becomes a Generalized Linear Model (GLM), which is a class of models that has been studied in great detail [41, 68–70]. In this section, we describe two such simplifications.

We can assume that the standard deviation is always proportional to the mean: $\boldsymbol{\sigma} = \sigma_0\,\boldsymbol{\mu}$. This assumes that the CV is constant through a high-frequency train, a coarse assumption given the large changes in CV observed experimentally [22, 54]. If for some reason an accurate reproduction of the changes in variability can be sacrificed, this simplification leads to interesting properties. In this case, no variance parameters are to be estimated apart from the scaling $\sigma_0$. There is thus a reduction in the number of parameters to be estimated. In addition, since the gamma distribution belongs to the exponential family and the mean is a linear-nonlinear function of the other parameters, we satisfy the requirements for GLMs. In some similar models, the likelihood function is convex [41], but since this is not the case in general [69], parameter inference must control for the robustness of solutions.

For the depressing synapses, the CV is increasing during a high-frequency train. This can be modeled by a constant standard deviation with a mean decreasing through the stimulus train. Similarly, for the facilitating synapses at normal extracellular calcium shown in Fig 2, the gradual decrease in CV can be explained by an approximately constant standard deviation, $\boldsymbol{\sigma} = \sigma_0$, and an increasing mean. Setting the variance to a constant again reduces the number of parameters to be estimated and recovers the necessary assumptions of a GLM.

## Relation to convolutional neural networks with dropout

A convolution followed by a nonlinear readout is also the central operation performed in convolutional neural networks (CNNs). Because this type of algorithm has been studied theoretically for its information processing capabilities and is associated with high performance in challenging tasks, we describe here one mapping of the biological models of information processing onto a model of the type used in artificial neural networks. Our main goal is to relate our SRP model with models in the machine learning literature.

CNNs are often used on images, and such inputs are conceived in two spatial dimensions but CNNs on data with a single temporal dimensions offer a more straightforward relationship with the properties of short-term plasticity. Such CNNs consider an input arranged as a one-dimensional array $\mathbf{x}$, which is convolved with a bank of kernels $\{\mathbf{k}_i\}$ and readout through a nonlinearity $f$ to generate the activity of the first layer of 'hidden' neural units

$$h_t^{(i)} = f(\mathbf{k}_i^T(\mathbf{m} \odot \mathbf{x}_{t:t+K})) \tag{7}$$

where $K_i$ is the length of the $i$th kernel in the bank. The convolution is here implied by the matrix multiplication, which applies to a section of the input and is shifted with index $t$. The bank of kernels extracts a number of different features at that neural network layer.

In Eq 7, we have added a mask $\mathbf{m}$ which operates on the input with the Hadamard product ($\odot$). This mask is introduced to silence parts of the input, randomly and ensure that learning yields kernels robust to this type of noise, an approach called *dropout* [71]. It is made of samples from Bernoulli random variable normalized so that the average of $\mathbf{k}_i^T(\mathbf{m} \odot \mathbf{x}_{t:t+K})$ is $\mathbf{k}_i^T \odot \mathbf{x}_{t:t+K}$.

Although CNN architectures vary, the next layer may be that of a pooling operation

$$h_k^{(i)} = \frac{1}{Z} \sum_{t=k}^{k+Z} h_t^{(i)}$$

where $Z$ is the pooling size, in number of time steps. Then these activities reach a readout layer for predicting higher-order features of the input

$$y_k = f(\mathbf{w}^T \mathbf{h}_k) \tag{8}$$

where the vector $\mathbf{w}$ weighs the pooled activities associated with the different kernels in the filter bank. By optimizing the kernels $\mathbf{k}$ and weights $\mathbf{w}$, similar CNNs have been trained to classify images [71, 72] as well as sounds [73, 74].

In a synapse with STP, the discretized efficacy train of the $i$th afferent, $\mathbf{e}_t^{(i)}$, results from a convolution and a nonlinear readout of the discretized spike train $\mathbf{s}_t^{(i)}$

$$e_t^{(i)} = f\left(\mathbf{k}_i^T \mathbf{s}_{t:t+K}^{(i)}\right) \mathbf{s}_t^{(i)}, \tag{9}$$

which maps to a discretized version of the continuous time SRP model in Eqs 2 and 3. By comparing with Eq 7, this equation (Eq 9) makes clear the parallel with a convolutional layer. Here, the spike train is conceived as a stochastic random variable sampling a potential [34, 48, 49]. Thus, the stochastic spike train is analogous to the dropout mask, $\mathbf{m}$. The efficacy train triggers PSCs, which are pooling the efficacy train on the PSC: $e_k^{(i)} = \sum_{t=k}^{k+Z} \epsilon_{t-k} e_t^{(i)}$, where $\epsilon_i$ is a discretized and normalized PSC. Then, different synaptic afferents, with possibly different efficacy kernels (Fig 9), are combined with their relative synaptic weights before taking a nonlinear readout at the cell body [34, 49] or the dendrites [75] to give rise to an instantaneous firing rate $\rho_t$:

$$\rho_t = f(\mathbf{w}^T \mathbf{e}_t) \tag{10}$$

This equation corresponds to the fully connected layer that followed a pooling operation, Eq 8. Together, we find a striking parallel between the formalism developed here to describe STP and that of an artificial neural network by ascribing a number of biological quantities to concepts in artificial intelligence. A number of these parallels have been made in the literature: Stochastic firing as a dropout mechanism [71], PSP as a pooling operation in time, and synaptic weights as connection weights. In addition, we find that the SRP model introduces a bank of temporal kernels with their nonlinear readout, which makes explicit that single neurons act as multi-layer neural network even in the absence of dendritic processing.

## Discussion

The linear-nonlinear framework has been able to capture core elements of subcellular [47], cellular [34, 36, 37, 76] and network signalling. We have shown that the same framework aptly captures synaptic dynamics. In the SRP framework, activity-dependent changes in efficacy are captured by an efficacy kernel. We have shown that switching the polarity of the kernel

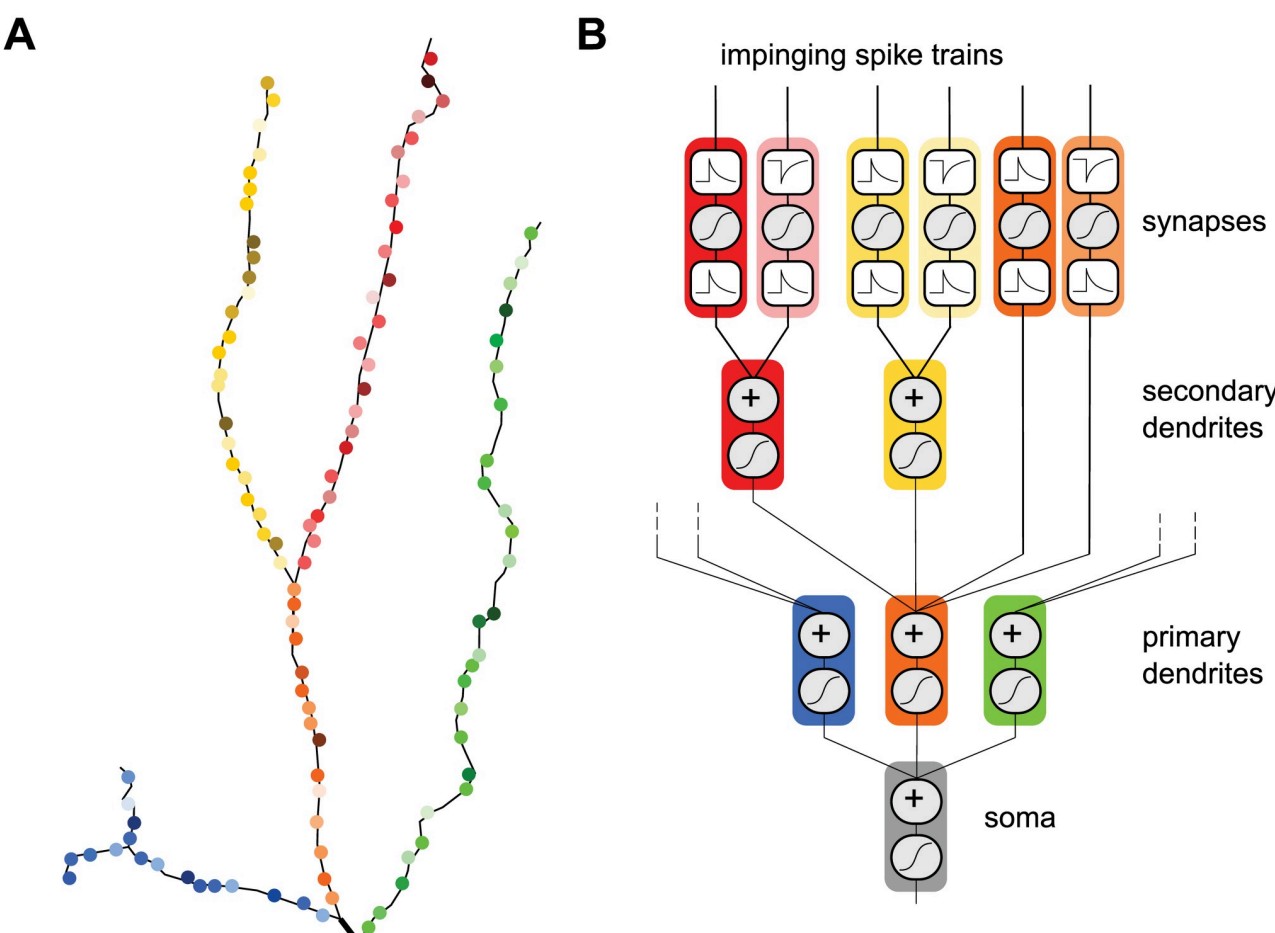

**Fig 9. A synaptic contribution to the hierarchy of linear-nonlinear computations. A** Synapses distributed on primary (orange, blue and green) and secondary (yellow and red) dendrites may have different synaptic properties (different color tints). **B** Each synapse is characterized by two kernels separated by a nonlinear sampling operation. 1) A pre-synaptic convolution kernel regulates synaptic dynamics. 2) A post-synaptic convolution kernel regulates the shape of the post-synaptic potential locally. The post-synaptic potentials from different synapses are summed within each dendritic compartment, forming a processing hierarchy converging to the soma.

captures whether STD or STF is observed. Extending previous work at ribbon synapses [77], we have shown that the modelling framework captures multiple experimental features of synaptic dynamics. The SRP model presents three sources of added flexibility with respect to the well-established TM model: 1) an efficacy kernel with an arbitrary number of timescales, 2) a nonlinear readout with both supra- and sub-linear regimes, and 3) an additional kernel allowing for independent dynamics of variability. The model successfully predicted experimentally recorded synaptic responses to various stimulation protocols, and reproduced the changes in variability incurred by changing the levels of extracellular calcium. The framework can also naturally capture long-lasting effects such as post-burst facilitation. Finally, by considering the dynamics of stochastic properties, a maximum likelihood approach can estimate model parameters from complex, time-limited, and physiological stimulation patterns. The added flexibility and the efficient inference are of interest to large scale characterization of synaptic dynamics [78] as well as the understanding information processing in neural networks [15, 79].

When summarizing dynamic properties with two time-dependent functions we called kernels, we were compelled to ask what was their biophysical implementation? By analogy with characterization of neuronal excitability, the answer is likely to involve a mixture of independent mechanisms. The membrane kernel, for instance, depends on membrane resistance and membrane capacitance, but also the density of low-threshold channels, such as A- and H-type currents. Similarly, the efficacy kernel is likely to reflect residual presynaptic calcium concentration and the changing size of the readily releasable pool [31] but also many other possible mechanisms. Determining the relative importance of these processes, however, is not possible with the methodology described here. This could be achieved only with a combination of experiments aimed at isolating independent mechanisms and a detailed biophysical model, at the cost of constructing a model with reduced predictive power. Our modeling framework is not presented as a tool for identifying molecular mechanisms, but rather as one for characterization, network simulation, and theoretical analysis [25, 80, 81] of the diversity of synaptic dynamics across signalling pathways [17], cell types [14, 50] or subcellular compartments [82].

There remain limitations to this approach such as the choice of a gamma distribution of release sizes. Formally, this modeling choice means that we have replaced release failures with small to very small releases. In other terms, whereas the presence of release failures makes a bimodal or multimodal distribution of amplitudes, the SRP model assumes that the distribution of evoked amplitudes is unimodal. Nonetheless, recent work has shown that the release size distribution appears unimodal despite being generated by multiple modes [56]. We have argued that for the small vesicle sizes at central synapses, quantal peaks are smeared by quantal variability [56]. When considering electrophysiological preparations where multiple synapses are simultaneously activated [27, 60, 83], the diversity of synaptic weights will strengthen further the assumption for a gamma-distributed, right-skewed and unimodal distribution.

Another related question is that, having explored various monotonic progressions of variability, will the model capture a non-monotonic progression? This case is relevant because the random and equally likely release of a number of vesicles will give rise to a non-monotonic progression of variability when release probability is changing over a larger range. For instance, in a facilitating synapse where multiple release sites increase an initially low release probability through a high-frequency train, the variability will first increase and then decrease. This convex, non-monotonic progression arises from the fact that variability is at its lowest point either when release probability is zero or when it is one. Given the mathematical features of the model, it may be possible to generate such a non-monotonic progression of variability with a biphasic $\sigma$-kernel.

Previous modeling and experimental work has established that dendritic integration can follow a hierarchy of linear-nonlinear processing steps [47, 75, 84]. Subcellular compartments filter and sum synaptic inputs through an integration kernel encapsulating local passive and quasi-active properties. Active properties are responsible for a static nonlinear readout and for communication toward the cell body. Much in the same spirit, the work presented here extends this model by one layer, where presynaptic spikes first pass through a linear-nonlinear step before entering dendrites (Fig 9). Since synapses at different locations or from different pathways may have different synaptic dynamics [17, 82], and since spiking neural codes can multiplex streams of information [8, 85, 86], these synaptic properties have the capability to extract different streams of information from multiple pathways and to process these possibly independent signals in segregated compartments.

The structure of information processing arising from this picture bears a striking resemblance with multi-layer convolutional neural networks [87, 88]. But it should be noted that the convolution takes place along the temporal dimension instead of the spatial dimension of many neural network applications. Yet, this algorithmic similarity suggests that a

linear-nonlinear structure of synaptic processing capabilities is shared between neural and neuronal networks. Whether the STP is controlled by genes [89], activity-dependent plasticity [90, 91], retrograde signalling [92], or neuromodulation [93, 94], a particular choice of efficacy kernels, when combined with a nonlinear readout, can optimize information processing as in Refs. [8, 95, 96].

## Methods

All models, numerical simulations and parameter inference procedures were implemented in Python using the SciPy and NumPy packages and are publicly available on GitHub (https://github.com/nauralcodinglab/srplasticity).

### Tsodyks-Markam model and its modifications

The Tsodyks-Markram (TM) model was first presented in 1997 [24] as a phenomenological model of depressing synapses between cortical pyramidal neurons and was quickly extended to account for short-term facilitating synapses [11, 50]. In the TM model, the normalized PSC amplitude $\mu_n$ at a synapse caused by spike $n$ of a presynaptic spike train is defined as:

$$\mu_n = R_n u_n, \tag{11}$$

where two factors $u_n$ and $R_n$ describe the utilized and recovered efficacy of the synapse, respectively. The temporal evolution of these variables are described by the following ordinary differential equations:

$$\frac{dR(t)}{dt} = \frac{1 - R(t)}{\tau_R} - u(t^-)R(t^-)S(t) \tag{12}$$

$$\frac{du(t)}{dt} = \frac{U - u(t)}{\tau_u} + f[1 - u(t^-)]S(t), \tag{13}$$

where $f$ is the facilitation constant, $\tau_u$ the facilitation time scale, $U$ the baseline efficacy and $\tau_R$ the depression timescale. The spike-dependent changes in $R$ and $u$ are implemented by the Dirac delta function within the spike train $S(t)$. The notation $t^-$ indicates that the function should be evaluated as the limit approaching the spike times from below.

In the TM model, facilitation is modelled as spike-dependent increases in the utilized efficacy $u$. Immediately after each spike, the efficacy increases by $f(1 - u(t^-))$. This efficacy jump depends on a facilitation constant $f$ and on the efficacy immediately before the spike $u(t^-)$. Therefore, as $u$ increases during a spike train, the spike-dependent 'jump' decreases for each subsequent spike. As a consequence, TM models of facilitating synapses are limited to a logarithmically saturating—that is, sublinear—facilitation.

To allow supralinear facilitation, we introduce a small change in the spike-dependent increase of factor $u$:

$$\frac{du(t)}{dt} = \frac{U - u(t)}{\tau_u} + u(t^-)f[1 - u(t^-)]\delta(t - t_S). \tag{14}$$

In this new model, given a presynaptic spike train at constant frequency, the size of the spike-dependent jump $u(t^-)f[1 - u(t^-)]$ saturates logarithmically for $u > 0.5$ but is increasing exponentially while $u < 0.5$. Thus this model provides supralinear facilitation in the low efficacy regimen, and it switches to sublinear facilitation for larger efficacies.

These models can be integrated between two spikes $n$ and $n + 1$, separated by time $\Delta t$ to speed up the numerical implementation [50]. For the classic TM model we have

$$R_{n+1} = 1 - [1 - R_n(1 - u_n)] \exp\left(-\frac{\Delta t}{\tau_R}\right) \tag{15}$$

$$u_{n+1} = U + [u_n + f(1 - u_n) - U] \exp\left(-\frac{\Delta t}{\tau_u}\right) \tag{16}$$

Similarly, the generalized model introduced in this work can be integrated between spikes:

$$u_{n+1} = U + [u_n + f(1 - u_n)u_n - U] \exp\left(-\frac{\Delta t}{\tau_u}\right) \tag{17}$$

Where $u_n^+ = u_n + f(1 - u_n)u_n$ is the value of $u$ after the spike-dependent increase following the $n^{th}$ spike. In both models, at time $t = 0$, we assume no previous activation, therefore $R_0 = 1$ and $u_0 = U$.

## Statistical inference

To extract the properties of the model from experimental data, we developed a maximum likelihood approach. Given a set of amplitudes $\mathbf{y} = \{y_1, y_2, \ldots, y_i, \ldots, y_n\}$ resulting from a stimulation spike-train $S$, we want to find the parameters $\theta$ that maximize the likelihood $p(\mathbf{y}|S, \theta)$. For this, as discussed in the body of the manuscript, we used a reparameterized gamma distribution such that the shape parameter and scale parameter are written in terms of the mean, $\mu = \gamma\lambda$, and standard deviation, $\sigma = \sqrt{\gamma}\lambda$. This results in a shape parameter: $\gamma = \mu^2/\sigma^2$, and scale parameter: $\lambda = \sigma^2/\mu$. The gamma distribution is then given by:

$$g(y|\mu, \sigma) = \frac{e^{\frac{-y\mu}{\sigma^2}} y^{(\mu^2/\sigma^2 - 1)}}{\Gamma\left(\frac{\mu^2}{\sigma^2}\right)\left(\frac{\sigma^2}{\mu}\right)^{\mu^2/\sigma^2}} \tag{18}$$

Thus, for the mathematical model presented here, the negative log-likelihood (NLL) is:

$$NLL(\mathbf{y}|S, \theta) = \sum_i \left[ \frac{y_i\mu_i}{\sigma_i^2} - \left(\frac{\mu_i^2}{\sigma_i^2} - 1\right) \ln\left(\frac{y_i\mu_i}{\sigma_i^2}\right) + \ln\left(\frac{\Gamma\left(\frac{\mu_i^2}{\sigma_i^2}\right)\sigma_i^2}{\mu_i}\right) \right] \tag{19}$$

where $\mu_i$ and $\sigma_i$ are shorthand for efficacy and standard deviation at the ith spike time: $\mu_i = \mu(t_i)$, $\sigma_i = \sigma(t_i)$, that is, the elements of the vectors $\boldsymbol{\mu}$ and $\boldsymbol{\sigma}$.

We parametrized the time-dependent standard deviation and mean of the gamma distribution by expanding the filters $\mathbf{k}_\mu$ and $\mathbf{k}_\sigma$ in a linear combination of nonlinear basis: $k_\mu(t) = \sum_l a_l h_l(t)$, and $k_\sigma(t) = \sum_m c_m h_m(t)$. Typical choices for such nonlinear basis are raised cosine [32], splines [66, 67], rectangular [97] or exponential decays [34]. In counterpart to the numerical simulations where the kernels are made of a combination of exponential functions with different decay time constants, we have used this choice of basis functions.

In this framework, hyper-parameters are the choice of the number of basis functions, $l \in [0, L]$ and $m \in [0, M]$, as well as the decay timescale for each basis function $h_l(t) = \Theta(t)e^{-t/\tau_l}/\tau_l$, where $\Theta(t)$ is a Heaviside function. Free parameters are the amplitude of the basis functions $\{a_l\}, \{c_m\}$ and the scaling factor $\sigma_0$. By choosing hyper-parameters *a priori*, the modeller must choose a number of bases that is neither too big to cause overfitting, nor too small to cause model rigidity. The choice of time constant is made to tile exhaustively the range of

physiologically relevant time scales. It is important to note that, because a combination of exponential basis functions can be used to capture a decay time scale absent from the set of $\tau$ hyper-parameters, the choice of $\tau$ does not specify the time scale of synaptic dynamics. The time-scale will be determined by inferring the relative amplitude of the basis functions. We can label the baseline parameter as the coefficient regulating the amplitude of a constant basis function, such that $a_0 = bh_0(t) = b_\mu$ and $c_0 = b_\sigma h_0(t) = b_\sigma$. There are thus $L + 1 + M + 2$ free parameters in total:

$$\theta = \{a_0, ..., a_L, \sigma_0, c_0, ..., c_M\}$$

To perform parameter inference, we first filtered the data using the set of basis functions and stored the filtered spike train just before each spike in a matrix. Each row of the matrix corresponds to an individual basis function, and each column corresponds to spike timings. The matrix, $X$, thus stores the result of the convolution between the various basis function (rows) and the spike train at the time of the various spikes (columns).

For simplicity, it is convenient to take the same choice of basis functions for the efficacy and the variance kernel. The amplitudes are expressed in a vector $\theta_\mu = \{a_0, \ldots, a_L\}$, for the efficacy kernel, and $\theta_\sigma = \{c_0, \ldots, c_M\}$ for the variance kernel. Using this matrix notation, the linear combination is expressed as a matrix multiplication:

$$\boldsymbol{\mu} = \frac{1}{f(a_0)} f\left(X^T \boldsymbol{\theta}_\mu\right)$$

$$\boldsymbol{\sigma} = \sigma_0 f\left(X^T \boldsymbol{\theta}_\sigma\right)$$

where $\boldsymbol{\mu}$ and $\boldsymbol{\sigma}$ have length $n$ and can be used to evaluate the NLL according to Eq 19 and $f(.)$ denotes the nonlinear (sigmoidal) readout. Performing a grid search of the parameter space around initialized parameter values, we can obtain the landscape for the function and ascertain the presence of convexity (see Fig 7). The inferred parameters will then be the set of $\theta_\mu$ and $\theta_\sigma$ minimizing the NLL over the training set.

### Fitting models to surrogate and experimental data

**SRP model.** Minimization of the NLL across the training set was performed using a limited-memory Broyden–Fletcher–Goldfarb–Shanno algorithm with bounded parameter constrains (L-BFGS-B). For the optimization on surrogate data in Fig 7, we chose an initial parameter estimate close to the true parameters. The resulting optimal parameter set provides an estimate of how close the NLL minimum is to the true parameter set for different numbers of training spikes. To avoid local minima when fitting the model to the experimental data in Fig 8, we we combined the L-BFGS-B minimization algorithm with a multistart procedure. We implemented a coarse grid search across the parameter space to generate a total of 256 equally spaced starting points for the optimization algorithm and kept the parameter set that yielded the minimum NLL across all converged optimizations.

**TM model.** To fit the TM model to experimental data in Fig 8, we implemented a thorough grid search across the 4-dimensional parameter space, probing 1 million parameter combinations. We kept the parameter set that minimized the mean squared error (MSE) across the training data.

## Acknowledgments

We thank Alexandre Payeur, Ezekiel Williams, Anup Pilail, Emerson Harkin and Jean-Claude Béïque for helpful comments.

## Author Contributions

**Conceptualization:** Richard Naud.

**Data curation:** Julian Rossbroich.

**Formal analysis:** Richard Naud.

**Funding acquisition:** Katalin Tóth, Richard Naud.

**Investigation:** Julian Rossbroich, Daniel Trotter, Richard Naud.

**Methodology:** Julian Rossbroich, Daniel Trotter, Richard Naud.

**Resources:** Katalin Tóth.

**Software:** Julian Rossbroich, John Beninger.

**Supervision:** Katalin Tóth, Richard Naud.

**Validation:** Julian Rossbroich, Katalin Tóth.

**Writing – original draft:** Julian Rossbroich.

**Writing – review & editing:** Julian Rossbroich, Daniel Trotter, John Beninger, Richard Naud.

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
