## [Decision Letter · Decision Letter 0]

27 Sep 2020

Dear Dr. Naud,

Thank you very much for submitting your manuscript "Synaptic dynamics as convolutional units" for consideration at PLOS Computational Biology.

As with all papers reviewed by the journal, your manuscript was reviewed by members of the editorial board and by several independent reviewers. In light of the reviews (below this email), we would like to invite the resubmission of a significantly-revised version that takes into account the reviewers' comments.

We cannot make any decision about publication until we have seen the revised manuscript and your response to the reviewers' comments. Your revised manuscript is also likely to be sent to reviewers for further evaluation.

Sincerely,

Boris S. Gutkin

Associate Editor

PLOS Computational Biology

Samuel Gershman

Deputy Editor

PLOS Computational Biology

Reviewer's Responses to Questions

**Comments to the Authors:**

Reviewer #1: Here the authors present a flexible, linear-nonlinear model of synaptic dynamics. They show how this model can describe two phenomena that are somewhat of a challenge to explain with more mechanistic models: sublinear and supralinear facilitation and facilitation latency. Overall, this is a clearly written paper and the shared code makes it more likely that other researchers can verify and build on these results. This work will certainly be interesting to many computational and experimental neuroscientists.

Major Issues:

1) The demonstrations for how the model can describe basic STD and STF, sub/supralinear facilitation, and facilitation latency are all clear and help motivate the model. However, it is less clear that the model actually outperforms previous models on real responses either to controlled stimulation in vitro or during natural activity in vivo. Adding model comparison along these lines (even for a few synapses) would help strengthen the paper (e.g. with publicly available data https://portal.brain-map.org/explore/connectivity/synaptic-physiology and comparing using publicly available code from previous models https://senselab.med.yale.edu/ModelDB/showmodel.cshtml?model=149914).

2) Adding too much flexibility could mean that the model may over-fit synaptic dynamics (and more easily be influenced by long-term plasticity too). Some discussion or analysis of this issue and the potential effect on generalization would be valuable.

3) The justification for the variance kernel is not entirely convincing. Having a highly flexible model of heteroskedasticity is nice in some ways, but the origins of variability will be less clear. For example, it seems like a fixed mean-variance relationship (approximately Binomial) could explain the changes in CV in Fig 2/6. This structure (where \\sigma_i = f(\\mu_i)) also seems relatively straightforward within this model framework. At the same time, both the kernel and fixed mean-variance models would miss the fact that part of the hetereoskedasticity is due to sequential sampling where y_j and y_j-1 are not independent given S. Some additional discussion of when and why you would want *such* a flexible model of variance would be helpful.

4) Relation to Convolutional Neural Networks – It’s not clear to me exactly what this section adds to the results beyond an analogy to deep learning. It seems that this is less of a “result” and more of a discussion point, one that is already made nicely in the paragraph around line 438. Is there something that I’m missing? Some novel way of thinking about synaptic dynamics across multiple inputs? If not it just seems to point out a possible extension of the model from Ujfalussy et al., and I would suggest removing or deemphasizing it in the results.

Minor Issues:

The title could be more specific.

Abstract – it would be helpful to clarify that these are short-term dynamics

Line 11: “the connectome” might be not be familiar to all readers. Could just say “anatomical/structural connectivity.”

Line 52 Typo: “can be inferred accurately with limited amount of experimental data”… should be “a limited” or “amounts”

Line 53 – “Our work also makes explicit that synaptic dynamics extend the information processing of dendritic integration by adding another layer of convolution combined with nonlinear readout…” The way this is worded makes it sound like a fact rather than a way of conceptualizing/modeling.

Fig 1 caption – could change “impulse response change in efficacy” to “efficacy kernel” for consistency.

“Sublinear and Supralinear Facilitation” ~line 135 – adding a brief overview of the TM model would be helpful here for readers who might not remember it exactly – e.g. state variables, parameters.

Line 167 – This paragraph is a bit unclear. Rather than “this variable”, it might be better to say “the baseline facilitation parameter”, throughout.

Fig 3 B/D/F – it might be helpful to normalize or change the axes on these plots so that B/D don’t appear range-restricted compared to F.

Line 195 Typo: “burst of action potentials”

Line 216 Typo: “Fig 3A” should be “Fig 2A”?

Line 291 Typo: “posterior distributions”

Line 342: “when the proof given by Paninski applies”… it’s not obvious to me that it does apply with a logistic nonlinearity and gamma noise model. It would be helpful to specify whether or not it does with the f(.) assumed in this paper. It also may be useful to point out in this section that mapping to a GLM does not guarantee identifiability/convergence (Zhao and Iyengar, Neural Comp 2010) or give any protection against model misspecification (Stevenson, Neural Comp 2018).

Line 453: “…this algorithmic similarity suggests that the linear-nonlinear structure of synaptic processing capabilities on neural and neuronal networks.” This sentence seems incomplete.

Line 457: “…can optimize information processing.” Not sure this claim is justified. Optimize how? What cost function?

Line 503: It would be helpful to spell out the gamma distribution (e.g. shape and scale parameters).

Line 518: Should include what nonlinearity was actually used for f(.)

Reviewer #2: Synaptic Dynamics as Convolutional Units

Julian Rossbroich, Daniel Trotter, Katalin Tóth, and Richard Naud

In this manuscript, the authors propose a novel framework to describe short-term synaptic dynamics such as short-term depression, facilitation and combinations of the two. The model is based on a linear-nonlinear operation. The deterministic model and its stochastic extension are shown to describe complex dynamics of the synaptic efficacy mean and variance observed in experiments. The manuscript proposes an noteworthy alternative to describe synaptic short-term dynamics compared to the standard Tsodyks-Markram model.

The aim of most of my comments is to better understand the model.

1. How well would the model perform when cross-validated over different sets of experiments, i.e., experiments with different presynaptic stimulation patterns? It would be curious to see how well the model matches with experimental data when fitted to one stimulation patterns and exposed to another.

Also, can the model be fitted to quantitatively reproduce the experimental data? The “Inference” section (starting on pg. 11) only unsatisfactorily answers this question. In particular, the data and model results shown in Fig. 2B and Fig 3F appear different in the sense that the efficacy at physiological calcium concentration appears to close to zero in the model. An overlay figure and a quantification comparing data and model would help to judge how well the model captures the data.

2. Why would the authors assume that the mono-exponential decay time is know in the inference method? This is hardly realistic when starting from experimental data with irregular stimulation patterns, for example. Why is the decay time not included in the inference demonstration?

3. The authors argue that the model does not require a change in its structure to capture sublinear, supralinear facilitation or delayed facilitation. The model is presented to be “more adaptable” compared to the Tsodyks-Markram model. I would argue that all complexity is put in the efficacy kernel which has to be adapted to capture the different short-term plasticity phenomenon. Difficult to judge which model is more flexible.

4. The introduction can do a better job in adequately justifying the model proposed here. In particular, the argument presented between lines 13 and 21 is reductionist. No experimentalist or modeler would claim that the full extent of short-term plasticity dynamics is captured by classifying synapses as short-term facilitating or depressing based on paired-pulse ratios. Furthermore, it remains unclear what is referred to as “complex STP dynamics”, or “complex synapses”. The short-comings of existing modeling approaches and the motivation for the modeling approach taken can certainly be presented in a more nuanced way.

5. Not being an expert in convolutional neural networks, I had problems to follow the related section on page 13 (l. 351-384). Maybe this part of the manuscript text could better tailored to a broader (non-CNN-expert) audience.

Other comments :

p. 4, l. 100 : “..., a nonlinear readout is potentially more apt .... “. Is that something the reader should be able to understand at this point?

p. 6, l.181 : Mention here that the CV dynamics will be considered later in the model framework. I was disappointed while reading that it was glanced over at this point of the manuscript.

Fig. 3 : Which panel in A and C corresponds to which case shown in B and D?

Fig. 4 : Panel D and E labels convey the same message and do not reflect what is shown in D.

p. 9, l. 216 : It should probably read “Fig. 2A” in the parenthesis.

Fig. 7 : It is not clear to me why C would indicate little correlation (“l.311”).? If D shows anti-correlation, C seems to hint to positive correlation.

**Have all data underlying the figures and results presented in the manuscript been provided?**

Reviewer #1: Yes

Reviewer #2: Yes

PLOS authors have the option to publish the peer review history of their article (what does this mean?). If published, this will include your full peer review and any attached files.

Reviewer #1: No

Reviewer #2: No
---

## [Decision Letter · Decision Letter 1]

25 Feb 2021

Dear Dr. Naud,

We are pleased to inform you that your manuscript 'Linear-Nonlinear Cascades Capture Synaptic Dynamics' has been provisionally accepted for publication in PLOS Computational Biology.

Best regards,

Boris S. Gutkin

Associate Editor

PLOS Computational Biology

Samuel Gershman

Deputy Editor

PLOS Computational Biology

Reviewer's Responses to Questions

**Comments to the Authors:**

Reviewer #1: The authors have addressed all of my concerns. Congratulations on a very nice paper.

Reviewer #2: The modifications have clarified the model and the results presented in the revised manuscript.

The additional section on fitting the model to mossy fiber synapse dynamics and performance comparison with the TS model provides a real added value to the investigations. It is possible to add the variability generated by the model to the comparison in 8G (similar to 8F)? The STD in the data exhibits large changes and it would be interesting to see the evolution of the model variability during the spike train.

Reviewer #3: This paper introduces a modelling framework for synaptic short term plasticity based on linear-nonlinear models, which can be fit to recordings of postsynaptic current responses during presynaptic stimulation. This approach is quite attractive as it allows constructing efficient models in a data-driven way. These models are much more efficient to simulate than the dynamical models usually used to describe this phenomenon, and may be useful for simulations or larger networks, or (as shown) multiple inputs into a single neuron. The paper shows this approach can be used to reproduce a number of experiments with specific short term dynamics. An extension which is perhaps somewhat less convincing (in terms of its ability to generalise) is the inclusion of a stochastic transmission model - at a synapse the variance depends both on release probability and vesicle availability, quantities this type of model does not provide directly. Yet the use of an additional kernel can capture changes in response variance during ongoing transmission. Overall this is interesting and relevant work, certainly appropriate for this journal.

I joined the review process after a first round of reviews, and see that extensive revisions were made to address a variety of comments. In particular the inclusion of the analysis of mossy fiber synapses have strengthened the paper considerably. In my opinion it is ready for publication. Moreover, the paper is very well written and the data well presented. I found a few minor typos, so I recommend double-checking the manuscript before the final submission.

**Have all data underlying the figures and results presented in the manuscript been provided?**

Reviewer #1: Yes

Reviewer #2: Yes

Reviewer #3: Yes

PLOS authors have the option to publish the peer review history of their article (what does this mean?). If published, this will include your full peer review and any attached files.

Reviewer #1: No

Reviewer #2: No

Reviewer #3: **Yes: **Matthias H. Hennig

---

## [Editor Report · Acceptance letter]

11 Mar 2021

PCOMPBIOL-D-20-00920R1 

Linear-Nonlinear Cascades Capture Synaptic Dynamics

Dear Dr Naud,

I am pleased to inform you that your manuscript has been formally accepted for publication in PLOS Computational Biology. Your manuscript is now with our production department and you will be notified of the publication date in due course.

With kind regards,

Alice Ellingham
